

# A novel one-step quick assay for detection of SARS-COV2 antibodies across mammalian species

Xianjin Zhou

Department of Psychiatry, University of California San Diego, La Jolla, California, United States

## ABSTRACT

Severe acute respiratory syndrome coronavirus 2 (SARS-CoV2) has so far infected almost a hundred of millions of people and caused more than a million of death across the world. Many serological tests have been developed to track down virus infection in community via identification of antibodies against SARS-CoV2 virus. However, the tests vary in sensitivity, specificity, complexity, and speed. Here, I developed a simple, one-step, quick test to detect antibodies against SARS-CoV2 N (scN) nucleocapsid protein via direct visualization of antigen-antibody reaction. A total of 40 serum samples of SARS-CoV2 patients were purchased from RayBiotech. A total of 50 pre-pandemic human serum samples from San Diego Blood Bank were used as negative controls. After performing the one-step quick test of these 90 serum samples, I found that 39 samples are positive for anti-scN antibodies. All of the 39 positives are from the 40 SARS-CoV2 patients, suggesting that the one-step test is more sensitive than the lateral flow immunoassay (LFIA), the most widely used rapid antibody test. None of the 50 pre-pandemic samples is positive for anti-scN antibodies, indicating that the one-step test has an excellent specificity. The one-step test takes only ~5 min to detect the antibodies; and 1 ml of *Escherichia coli* culture can produce reagent proteins sufficient for thousands of the tests. Since the one-step test does not need a secondary antibody, it can be used as a universal test for anti-scN antibodies across different mammalian species to track down both human infection and the animal reservoir of SARS-CoV2 virus.

## INTRODUCTION

Antibodies, the biomarkers for a variety of human diseases, particularly infectious diseases (*Peruski & Peruski, 2003*), can be detected by many laboratory immunoassays such as enzyme-linked immunosorbent assay (ELISA), Western blot, cell-based assays, immunohistochemistry, etc. The pandemic of coronavirus disease 2019 (COVID-19) caused by SARS-CoV2 virus creates a sense of urgency for the development of rapid antibody tests with a high sensitivity and specificity to track down virus infection in community and provide clinical point-of-care for patients. ELISA, lateral flow immunoassay (LFIA), and direct chemiluminescence immunoassay (CLIA) are three assays mostly used in serological tests of human SARS-CoV2 infection (*Espejo et al., 2020*).

Corresponding author
Xianjin Zhou, xzhou@ucsd.edu

These assays vary in sensitivity, specificity, complexity, and speed. LFIA is the fastest immunoassay that detects anti-scN antibodies in ~15 min (*Li et al., 2020*). However, LFIA specificity and sensitivity are often compromised by many factors, particularly the supporting nitrocellulose membrane and protein stability after drying (*Pavlova et al., 2020*). In fact, all immunoassays including ELISA and CLIA use solid support and protein coating, which makes these assays prone to non-specific binding of antibodies (*Terato et al., 2016*). Here, I developed a novel, one-step, quick test to detect SARS-CoV2 antibodies by direct visualization of antigen-antibody reaction in solution rather than on solid support. This avoids non-specific background from both solid support and coated dried proteins. I selected the nucleocapsid protein as an antigen for assay development in order to compare the one-step quick test with the most widely used LFIA that detect antibodies against SARS-CoV2 nucleocapsid protein. The sensitivity of the one-step quick test is superior to LFIA in the same group of SARS-CoV2 patient sera.

## MATERIALS AND METHODS

### Production of scN-GFP in *Escherichia coli*

Nucleocapsid protein sequence of SARS-CoV2 virus was from NCBI reference sequence database (Accession NP_828858). Gene encoding scN-GFP fusion protein with a 6His tag was synthesized and cloned into pET-21d vector. BL21(DE3)pLysS competent *Escherichia coli* cells were purchased from EMD (cat. 70236-3) for transformation of the plasmid. In brief, a single *E. coli* colony was inoculated in 2 ml of LB medium containing 100 μg/ml carbenicillin. After overnight shaking at 37 °C, the *E. coli* culture was diluted 1:10 with LB medium containing both 100 μg/ml carbenicillin and 0.2 mM IPTG. The diluted *E. coli* culture was vigorously shaked for 4 h at 37 °C to induce over-expression of scN-GFP proteins. After centrifugation, *E. coli* cell pellet was suspended in 1X PBS, 0.25 M NaCl, 1 mM PMSF, and then sonicated on ice. scN-GFP proteins in the supernatant were collected after centrifugation. HisPur Ni-NTA resins (cat. 88221; ThermoFisher Scientific, Waltham, MA, USA) were washed with 1XPBS before loaded with the scN-GFP containing supernatant. After 10X volume of washing with 1X PBS, 0.25 M NaCl, scN-GFP proteins were eluted with 1X PBS, 0.25 M NaCl, 120 mM imidazole.

### One-step test

Rabbit polyclonal antibodies against SARS-CoV2 N nucleocapsid proteins were purchased from SinoBiological (cat. 40588-T62). Protein A/G/L was purchased from Novus Biologicals (NBP2-34985) and diluted to 1 ug/ul with antibody diluent solution (S080983-2; DAKO, Denmark). Protein A/G/L tagged with 6His was also over-expressed in *E. coli* and purified in my laboratory. The purified scN-GFP proteins were diluted with 1X PBS, 0.25 M NaCl for use. One μl of serum was mixed with 3 μl of the diluted scN-GFP and 1 ul of protein A/G/L (1 ug/ul). After incubation for 5 min at room temperature, antigen-antibody aggregates of the one-step test were examined for GFP green fluorescence using microscope EVOS FL (ThermoFisher Scientific, Waltham, MA, USA).

## Human serum samples

Forty serum samples from SARS-CoV2 patients diagnosed by PCR or antigen tests were purchased from RayBiotech (Supplemental Data 1). Blood from 37 patients were drawn 33–35 days after the diagnostic test. Blood of patient A11, A18, and B7 were drawn 64, 25, and 18 days after the test, respectively. Fifty pre-pandemic human serum samples were purchased from San Diego Blood Bank (Supplemental Data 2). The studies were approved by Human Research Protections Program at University of California San Diego.

## RESULTS

It is difficult to directly visualize single antigen-antibody molecule interaction for detection of antibodies. I therefore propose to aggregate millions of antigen-antibody molecules together to make the antigen-antibody reaction instantly visible. Protein A/G/L consists of 5 IgG-binding regions of protein A, 2 IgG-binding regions of protein G, and 5 light chain-binding regions of protein L. In a mixture of antibodies and fluorescence-labeled antigens, addition of protein A/G/L will instantly cross-link antigen-antibody molecules into large aggregates by binding heavy and/or light chain of antibodies (Figs. 1A–1B). Such large aggregates will emit strong fluorescence. As a proof-of-concept, SARS-CoV2 N nucleocapsid protein (scN) was fused to green fluorescence protein (GFP) with a 6His tag, over-expressed in BL21(DE3) *E. coli* cells, and purified with Ni-NTA resins. Rabbit polyclonal antibodies against SARS-CoV2 N nucleocapsid protein (cat. 40588-T62; SinoBiological, Beijing, China) were diluted in human serum to a concentration of 100 ng/ul, 10 ng/ul, 1 ng/ul, respectively. The scN-GFP proteins were diluted at ~30 ng/ul and incubated with the diluted rabbit antibodies and protein A/G/L (Figs. 1C–1E). After 5 min incubation at room temperature, the reactions were loaded into capillary slides (cat. 76237-746; VWR, Radnor, Pennsylvania, USA) for examination. As expected, antigen-antibody aggregates emit strong green fluorescence with an input of 100 ng of the rabbit antibodies. The limit of detection appears to be ~10 ng of the rabbit antibodies. To examine non-specific background of antigen-antibody aggregates, I incubated a mixture of control human serum, scN-GFP, and protein A/G/L for 5 days at 4 °C (Figs. 1F–1G). No fluorescence background was observed in the antibody aggregates from the control human serum, whereas strong fluorescence persists in the antigen-antibody aggregates from human serum containing the rabbit antibodies against SARS-CoV2 N proteins. These data suggested that the one-step quick test may have a high sensitivity and specificity.

To examine whether the one-step test can detect anti-scN antibodies in patients with SARS-CoV2 infection, I purchased 40 serum samples of patients diagnosed by PCR from RayBiotech (Supplemental Data 1). Patient blood were drawn 33–35 days after the diagnostic test, except for 3 patients whose blood were drawn 64, 25, and 18 days after the test, respectively. These samples had been studied by RayBiotech for antibodies against SARS-CoV2 N proteins using LFIA rapid antibody test. Samples are deemed positive if either IgG or IgM is positive by LFIA, and 20 out of the 40 samples are positive for anti-scN antibodies. The other 20 samples are negative by LFIA. 50 pre-pandemic human serum samples from San Diego Blood Bank were used as negative controls (Supplemental Data 2).

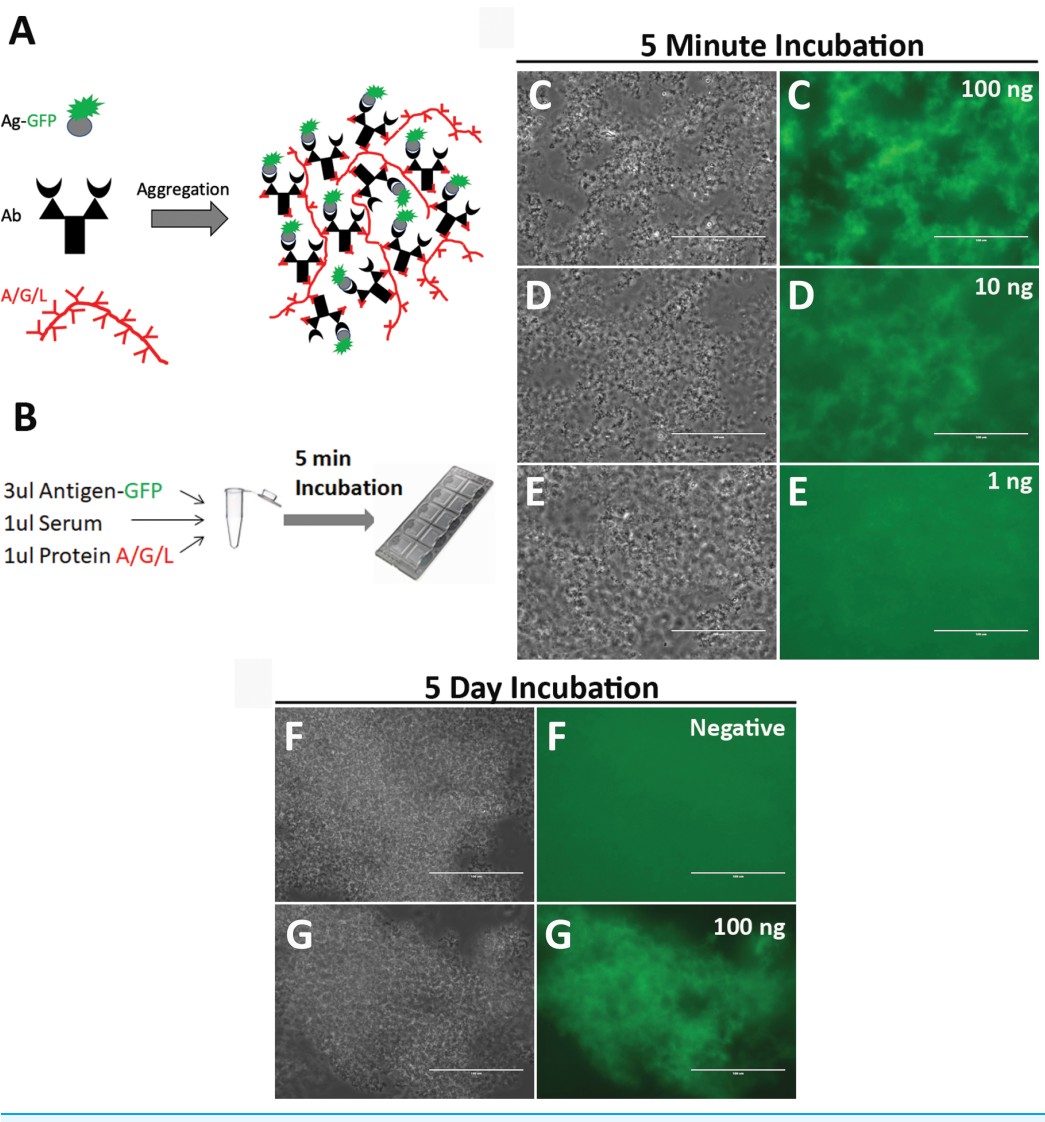

**Figure 1 Development of the one-step test for detection of antibodies against SARS-CoV2 N nucleocapsid protein.** (A) A strategy of direct visualization of antigen-antibody reaction via aggregation by protein A/G/L. Antigen is labeled with GFP. Protein A/G/L cross-links Ig Fc and/or light chain of all antibodies (IgG, IgM, IgA, IgE, and IgD) that bind antigen-GFP to form high fluorescence aggregates. Aggregating fluorescent antigen-antibody complexes simultaneously depletes background fluorescence to achieve a high sensitivity. (B) One-Step assay. Rabbit polyclonal antibodies against SARS-CoV2 N nucleocapsid proteins were diluted in human serum. Different amount of the rabbit antibodies (100 ng (C), 10 ng (D), 1 ng (E)) was incubated with the scN-GFP fusion proteins for 5 min to examine the detection limit of the one-step test. Blackwhite images show antigen-antibody aggregates, and green fluorescence of the aggregates is shown side by side. Bar: 100 um. Non-specific background of the antigen-antibody aggregates was examined after 5 day incubation of reactions at 4 °C. (F) A pre-pandemic human serum was used as a negative control. (G) A positive control was human serum containing 100 ng of the rabbit antibodies against SARS-CoV2 N nucleocapsid protein.

The one-step test was conducted to examine anti-scN antibodies in all of the 90 serum samples. A total of 39 positive serum samples were identified, and all of them came from the 40 SARS-CoV2 patients. None of the 50 pre-pandemic serum samples is positive for

| Controls | Patient Group A | Patient Group B |
|---|---|---|

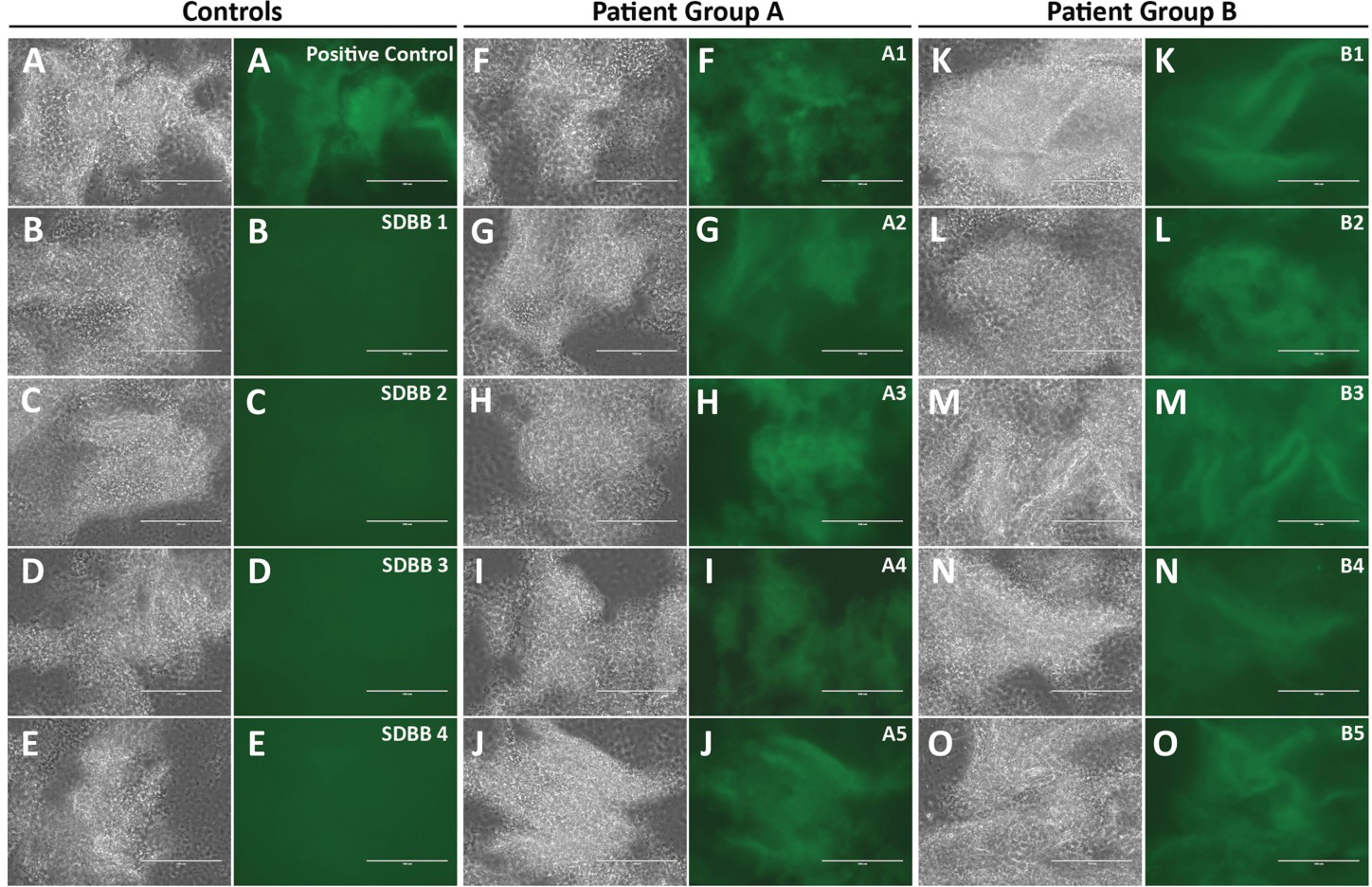

**Figure 2 Sensitivity and specificity of the one-step test in patient serum samples.** (A) The positive control was human serum containing 100 ng of the rabbit antibodies against SARS-CoV2 N nucleocapsid protein. The negative controls are 4 representative samples (SDBB1-4 (B–E)) from the 50 pre-pandemic human serum samples from San Diego Blood Bank (SDBB). Patient Group A has 20 SARS-CoV2 patient sera deemed positive (either IgG or IgM positive) by LFIA conducted by RayBiotech. All of the 20 patient samples are positive for anti-scN antibodies by the one-step test. (F–J) 5 representatives of the 20 positive samples are shown. Patient Group B has 20 SARS-CoV2 patient serum samples deemed negative (both IgG and IgM are negative) by LFIA conducted by RayBiotech. A total of 19 out of the 20 LFIA negative samples are positive for anti-scN antibodies by the one-step test. (K–O) 5 representatives of the 19 positive samples are shown.

anti-scN antibodies (Figs. 2A–2E, Controls). All of the 20 patient samples deemed positive by LFIA are also identified as positive by the one-step test (Figs. 2F–2J, Group A). Out of the 20 patient samples deemed negative by LFIA, 19 samples were identified as positive by the one-step test (Figs. 2K–2O, Group B). Figure 2 shows images of some representative controls and patients. Images of the rest of the controls (Figs. S1–S5) and the patients (Figs. S6–S8) were provided as supplemental figures. The results are summarized in Table I. As expected, antibodies against scN can be detected in the patients' blood more than one month after diagnosis. How long the antibodies persist in patient blood remains to be studied. The negative one by the one-step test may have a very low level of anti-scN antibodies or did not develop anti-scN antibodies since the patient was also negative by PCR diagnosis (Supplemental Data 1). In this small cohort of human samples, the one-step

**Table 1 Comparison between LFIA rapid antibody test and the one-step test.**

| | RayBiotech patients | | SDBB pre-pandemic controls |
| --- | --- | --- | --- |
| | Group A ($n$ = 20) | Group B ($n$ = 20) | ($n$ = 50) |
| Age (SD) | 64.3 (16.1) | 64.1 (17) | 52.5 (17.9) |
| Positives by LFIA | 20 | 0 | N/A |
| Positives by One-Step | 20 | 19 | 0 |

test has ~97% sensitivity and 100% specificity. In conclusion, the one-step test appears to be more sensitive than LFIA.

## DISCUSSION

Current immunoassays indirectly detect antibodies. It is difficult to "see" the contribution of non-specific background from solid surface and coated proteins in each reaction. The one-step test offers a new immunoassay to instantly visualize direct antigen-antibody reaction in solution rather than on solid support, avoiding all non-specific background except antibody cross-reactions. Reaction of antigens and antibodies in solution help maintain protein structure, which may contribute to its excellent specificity. Aggregating antigen-antibodies simultanouesly depletes background fluorescence, which enhances its detection sensitivity. The one-step test is conducted in a small reaction volume (5–6 ul) to achieve a high concentration of antibodies and antigens, which makes antigen-antibody reaction very fast (<5 min) for detection of antibodies. Since the one-step test does not need a secondary antibody and protein A/G/L binds antibodies from many different mammals, the exactly same test can be used to detect anti-scN antibodies across mammalian species. This could be particularly helpful for studies tracking down the animal reservoir of the SARS-CoV2 virus where secondary antibodies against the host animals are not available for classic immunoassays. Such tracking studies are important in preventing introduction and re-introduction of SARS-CoV2 from animals to human population.

LFIA is the most widely used rapid antibody test currently provided by Kroger and Ralph stores across the US. It detects anti-scN antibodies from past infection of SARS-CoV2 virus. The LFIA sensitivity varies from different studies, in part due to using different patient samples that have different levels of anti-scN antibodies. In this study, the one-step test is compared with LFIA in the same group of serum samples. The one-step test displayed a much higher sensitivity than LFIA (IgG and IgM combined sensitivity) in detection of anti-scN antibodies. Although LFIA specificity was not examined, the one-step test has 100% specificity in the 50 pre-pandemic human sera. Altogether, this study suggests that the one-step test be superior to LFIA in detection of anti-scN antibodies. However, the one-step test examines all anti-scN antibodies (IgG, IgM, IgA, IgE, and IgD), whereas LFIA can differentiate IgG and IgM anti-scN antibodies. All scN-based antibody tests may be complicated by cross-reactions with SARS-CoV1 nucleocapsid protein that have ~90% homology with the nucleocapsid protein of SARS-

CoV2 (*Dutta, Mazumdar & Gordy, 2020*). Their spike proteins, however, are less well conserved with 79.6% identity. Specific antigenic epitopes may be selected from the spike protein for the development of assays to differentiate their infections. It is unlikely that antibodies aganist nucleocapsid protein of SARS-CoV2 may cross-react with the nucleocapsid proteins of other flu-causing coronavirus due to their limited homology. Consistently, the one-step test did not detect any positive signals from the 50 pre-pandemic sera. More studies are needed to investigate cross-reactions between the anti-scN antibodies and antigens of other viruses.

A potential complication of the one-step test could arise from antibodies recognizing the GFP part of the fusion protein rather than the antigen part. However, I did not find any human sera binding GFP after screening ~200 human serum samples. Such false positives, if occur, can be readily ruled out by incubating the sera with GFP proteins. As an alternative approach to avoid potential cross-reaction of serum antibodies with GFP proteins, a small fluorescence dye such as fluorescein isothiocyanate (FITC) or Alexa Fluor rather than GFP could be conjugated to scN antigen for the one-step test. Different sera may have different levels of autofluorescence metabolites and/or proteins. The intensity of the green autofluorescence is, however, far less than the fluorescence intensity of the antigen-GFP proteins added into the assay. I only examine the increase of green fluorescence of large aggregated antibody-antigen-GFP complexes above the green fluorescence background that is mainly determined by free antigen-GFP proteins. Blood autofluorescence from soluble metabolites or proteins may slightly contribute to variations of background between individual samples. Since these soluble autofluorescence metabolites or proteins are not incorporated into the aggregates, they have little effect on the one-step test. I do not encounter a problem from blood autofluorescence in the assay. However, GFP tag can be replaced with a different fluorescence protein to avoid autofluorescence if needed.

An advantage of the one-step test is that both scN-GFP and protein A/G/L can be abundantly produced in *E. coli*. One ml of *E. coli* culture can generate reagent proteins enough for thousands of the one-step tests. Purified scN-GFP and protein A/G/L are stable for months in 4 °C refrigerator. Therefore, the one-step test could be particularly attractive for developing countries to produce sufficient diagnostic tests to track down SARS-CoV2 infection during the pandemic. In the future, development of smartphone fluorescence microscope (*Dai et al., 2019*) will enable the one-step test to be conducted by individuvals at home.

This study demonstrated the utilization of the one-step test in detection of anti-scN antibodies. Different antigens can however be labeled with fluorescence or other dyes for detection of their antibodies after aggregation. If SARS-CoV2 RBD is fused with GFP, the test would detect neutralizing antibodies against SARS-CoV2 virus. This could be particularly helpful for persons to monitor the levels of circulating SARS-CoV2 neutralizing antibodies after vaccination. More broadly, this strategy may be used to develop various one-step tests to detect different antibodies involved in a variety of human and animal diseases.

### Funding

The study is supported by R21 MH123705 (Xianjin Zhou). There was no additional external funding received for this study. The funders had no role in study design, data collection and analysis, decision to publish, or preparation of the manuscript.

### Grant Disclosures

The following grant information was disclosed by the authors:
R21 MH123705.

### Competing Interests

Xianjin Zhou is the inventor on a provisional patent filing by the University of California San Diego.

### Author Contributions

- Xianjin Zhou conceived and designed the experiments, performed the experiments, analyzed the data, prepared figures and/or tables, authored or reviewed drafts of the paper, and approved the final draft.

### Human Ethics

The following information was supplied relating to ethical approvals (i.e., approving body and any reference numbers):

The studies were approved by Human Research Protections Program at University of California San Diego.

### Patent Disclosures

The following patent dependencies were disclosed by the authors:

Methods for Detecting Antibodies (UCSD Ref. No. SD2021-104, Oct 10, 2020)

### Data Availability

The raw data are available in the Supplemental Files.

### Supplemental Information

Supplemental information for this article can be found online at http://dx.doi.org/10.7717/peerj.11381#supplemental-information.

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
