# Peer review of "A novel one-step quick assay for detection of SARS-COV2 antibodies across mammalian species"

_PeerJ, doi:10.7717/peerj.11381_

## Round 0.1 · original submission · Major Revisions

Dear Authors,

Your manuscript was evaluated by three reviewers. All reviewers have provided important suggestions to improve the current manuscript.

Sincerely,

Gunjan

Reviewer 1 ·

Basic reporting

The manuscript is well-written and clear to read.

It is recommended that the author discusses the rationale for study design and explains why the assay was developed against the nucleocaspsid region. Some additional background on the nucleocapsid vs spike region will add value to the general readers.

The author is suggested to include all the images as a supplementary file. Only representative images are currently included.

Experimental design

Some additional details regarding assay design can be included.

Validity of the findings

It is suggested to submit all the underlying data with regard to fluorescence images of negative controls and patient cohort.

Additional comments

In the manuscript submitted by Dr. Zhou, he describes the development of a non-amplification, solution based-testing strategy for antibodies against the SARS-CoV2 N nucleocapsid. This one-step rapid test is based on an agglutination reaction between antibodies in the patient sera with recombinant antigen fused to GFP, aided by protein A/G/L. This protein aggregates can be then visualized using a fluorescence microscope. The test was able to predict SARS-CoV2 infection in 39/40 patient samples and produced no false positivity among 50 pre-pandemic samples. The merits of this test surely is the time and ease of the testing. However, listed below are a few concerns regarding the manuscript in its current form.
1. The author is suggested to provide all the results from the pre-pandemic samples as well as the SARS-CoV2 patient cohort as supplemental data. It is recommended to show fluorescence images and a summary table.
2. Please discuss the time of blood draw in the methods and results section. The details of the time delay for the blood draw is mentioned in the supplemental table but not in the main manuscript. This is an important piece of information to suggest that the one-step rapid test can detect antibodies even after viral clearance. It is also suggested that the author discusses this point.
3. Is this assay amenable to quantification? Can the results provide more information than positive/negative status?
Here are a few minor suggestions for consideration and discussion:
1. The nucleocapsid region of SARS-CoV2 shares 90% homology with SARS-CoV1. Will this one-step test produce false positive results in patients who previously contracted SARS-CoV1?
2. Will the test produce false positive results if patients present with high antibody levels against another infection (non-SARS-CoV2 pathogen)? Will these other antibodies cause crosslinking among the protein A/G/L? The current negative samples from the San Diego Blood Bank may not help address this issue.

Reviewer 2 ·

Basic reporting

N/A

Experimental design

N/A

Validity of the findings

N/A

Additional comments

The focus of current manuscript by Dr. Xianjin Zhou is to develop a relatively quick and specific qualitative serological test to detect antibodies against SARS-CoV2. Currently multiple serological tests are available to detect antibodies against SARS-CoV2 that vary in specificity, sensitivity and the time. The author used GFP fused-antigen-antibody complex aggregation to visualize the test result by a fluorescence microscope. Following are the concerns..
1) Did the author try one-step method using only serum, or any other biological fluid?
2) Why did author incubate negative control mix for 5-days but not 5mins at room temp (Fig 1C) ?
3) Visualization of GFP signal is challenging because of autofluorescence. Can this test be replicated with any other fluorescence tag?
4) It was mentioned that 30µg/µl Antigen-GFP protein was used to test (Fig 1B). Is 30µg minimum amount needed? Did the author try varying concentrations of Ag-GFP mix with SARS-CoV2 positive sera / Rabbit polyclonal abs against SARS-CoV2 (in presence of A/G/L )? This should be tested as it helps to rule out the false positive outcome.
5) It is mentioned that the test volume is 5-6µl, what concentration of A/G/L was used? Does the test sensitivity depend on concentration of Ag-GFP and A/G/L ? or the amount of Abs present in sera? Varying signal intensities are present among patient samples in Fig2.
6) Relevant citations were missing in most part of the introduction.
7) Author should avoid text redundancy in methods and main text.

Reviewer 3 ·

Basic reporting

No comment

Experimental design

1. Author is requested to mention the materials and methods section in detail. The present information seems to be incomplete in assessing the reproducibility of the assay.
2. One-step test should be described in detail describing the concentration of antibodies, antigens used eliminating any assumptions during peer -review, and discussion.

Validity of the findings

1. It will be interesting to have the titer results shown with the antibody across a broad range in patients with varying infection severity.
2. Author should report when were the samples collected to detect the time point applicability of the assay.
3. Comparison should be made with rapid antibody detection tests available in the market or it should be highlighted in the discussion that how this test differs from the existing ones with respect to novelty, sensitivity, and specificity.

---

## Round 0.2 · Minor Revisions

Dear Authors,

Your manuscript was evaluated by three reviewers. One of the reviewers has suggested discussing autofluorescence as a limitation of current work.

Sincerely,

Gunjan

Reviewer 1 ·

Basic reporting

No comment

Experimental design

No comment

Validity of the findings

No comment

Additional comments

The author has systematically addressed the concerns raised in the first review report.

1. The author has updated the introduction to include the requested background information and rationale for study design.
2. The author has now provided image results for all the agglutination reactions as supporting data in supplemental files.
3. The methods and results section has been revised to include the blood draw details.
4. The author has discussed the issues of cross reactivity in the discussion section.

Overall, the author provided satisfactory responses to address the critiques.

Reviewer 2 ·

Basic reporting

NA

Experimental design

NA

Validity of the findings

NA

Additional comments

The author addressed all the concerns. However, I want to add the following comment.

It is not true that only formalin fixed tissues exhibit auto-fluorescence. Various substances, metabolites, and protein complexes of blood too exhibit auto-fluorescence (the auto-fluorescence properties of blood is being studied to use as diagnostic purposes). The author should discuss this aspect as a limitation or caution to avoid false positive result of the test (this is important as fluorescence intensity is the outcome of the test).

Reviewer 3 ·

Basic reporting

NA

Experimental design

The author has addressed all the concerns.

Validity of the findings

The author has addressed all the concerns.

---

## Round 0.3 · accepted · Accept

The author has discussed the limitation of their results.